# Impact of Soy β-Conglycinin Peptides on PCSK9 Protein Expression in HepG2 Cells

**DOI:** 10.3390/nu14010193

**Published:** 2021-12-31

**Authors:** Chiara Macchi, Maria Francesca Greco, Nicola Ferri, Paolo Magni, Anna Arnoldi, Alberto Corsini, Cesare R. Sirtori, Massimiliano Ruscica, Carmen Lammi

**Affiliations:** 1Dipartimento di Scienze Farmacologiche e Biomolecolari, Università degli Studi di Milano, 20133 Milan, Italy; chiara.macchi@unimi.it (C.M.); mariafrancesca.greco@unimi.it (M.F.G.); paolo.magni@unimi.it (P.M.); alberto.corsini@unimi.it (A.C.); cesare.sirtori@unimi.it (C.R.S.); 2Dipartimento di Medicina, Università degli Studi Padova, 35131 Padova, Italy; nicola.ferri@unipd.it; 3IRCCS MultiMedica, Sesto S. Giovanni, 20099 Milan, Italy; 4Dipartimento di Scienze Farmaceutiche, Università degli Studi di Milano, 20133 Milan, Italy; anna.arnoldi@unimi.it (A.A.); carmen.lammi@unimi.it (C.L.)

**Keywords:** soy peptides, PCSK9, LDL-R, nutraceuticals, HNF1α

## Abstract

Background: Dyslipidaemias, particularly elevated plasma low-density lipoprotein cholesterol (LDL-C) levels, are major risk factors for cardiovascular disease (CVD). Besides pharmacological approaches, a nutritional strategy for CVD prevention has gained increasing attention. Among functional foods, the hypocholesterolemic properties of soy are driven by a stimulation of LDL-receptor (LDL-R) activity. Aim: To characterize the effect of two soy peptides, namely, β-conglycinin-derived YVVNPDN**DEN** and YVVNPDN**NEN** on the expression of proprotein convertase subtilisin/kexin type 9 (PCSK9), one of the key-regulators of the LDL-R. Methods: PCSK9 promoter activity (luciferase assay), PCSK9 protein expression (WB) and secretion (ELISA), PCSK9 interaction with LDL-R (binding assay) and human HepG2 cells were the objects of this investigation. Results: Treatment with YVVNPDN**NEN** peptide has led to a rise in PCSK9 gene expression (90.8%) and transcriptional activity (86.4%), and to a decrement in PCSK9 intracellular and secreted protein (−42.9%) levels. YVVNPDN**NEN** peptide reduced the protein expression of transcriptional factor HNF1α. Most changes driven by YVVNPDN**DEN** peptide were not statistically significant. Neither peptide inhibited the PCSK9–LDLR interaction. Conclusions: Although sharing a common effect on LDL-R levels through the inhibition of 3-hydroxy-3-methylglutaryl CoA reductase activity, only the YVVNPDN**NEN** peptide has an additional mechanism via the downregulation of PCSK9 protein levels.

## 1. Introduction

Dyslipidaemias are plasma lipid alterations frequently associated with clinical conditions affecting the arterial system. In particular, elevated levels of low-density lipoprotein cholesterol (LDL-C) represent one of the major risk factors for cardiovascular disease (CVD) [1]. Thus, treatments aimed at reducing LDL-C unequivocally lead to fewer CV events [2].

In addition to pharmacological approaches, a nutritional strategy for CVD prevention has gained increasing attention [3]. Several “functional foods”, i.e., food items providing additional health benefit beyond energy production, have shown a clear activity on LDL-C levels. Among these are plant proteins, e.g., soybean proteins [4], providing an excellent source of protein, dietary fibres, and phytochemicals. The beneficial effects of plant compared to animal protein sources on cholesterol lowering have been confirmed by comparative studies on the intake of active daily protein doses, most typically of soy (daily range between 25 to 100 g) [5,6]. This approach leads to an LDL-C reduction between 3% and 20% [7].

The hypocholesterolemic properties of soy are attributed to a stimulation of LDL-receptor (LDL-R) activity [8,9]. This effect appears to be dependent on soy derived peptides, YVVNPDN**DEN** and YVVNPDN**NEN** [10,11]. The mechanism of these two absorbable peptides, belonging to the soybean β-conglycinin subfraction [12], has been unravelled with an integrated approach with in vitro, in silico, and cellular tools [11].

Both peptides reduce the 3-hydroxy-3-methylglutaryl CoA reductase (HMGCoAR) activity in vitro with a definite dose-response and IC_50_s equals to 150 and 200 μM, respectively. The key interactions stabilizing the putative complex between YVVNPDN**DEN** and the enzymatic activity of HMGCoAR have been confirmed by the ability of both peptides to inhibit the enzyme with a statin-like mechanism. Moreover, both YVVNPDN**DEN** and YVVNPDN**NEN** increase the LDL-R protein levels by the activation of the sterol regulatory-element-binding protein 2 (SREBP-2) transcription factor, thus raising the ability of human HepG2 cells to take up LDL from the extracellular compartment [11].

Proprotein convertase subtilisin/kexin 9 (PCSK9) is a liver-secreted plasma protein that regulates the number of cell-surface LDL receptors [13]. Genetic and interventional studies have demonstrated the CVD benefit of PCSK9 reduction [14]. In this area, several nutraceuticals have been shown to modulate PCSK9 expression, including berberine, curcumin, and various legumes [15]. PCSK9 synthesis is controlled at the transcriptional level by both SREBPs and by the hepatocyte nuclear factor1 alpha (HNF1α) [16,17]. On this matter, liver-specific knockdown of HNF1α significantly decreases serum PCSK9 expression [16]. HNF1α is a liver-enriched transcription factor which regulates many target genes in the liver and intestine [18].

Thus, the present study aimed to characterize the effects of soy peptides YVVNPDN**DEN** and YVVNPDN**NEN** on the regulation of PCSK9 expression. The PCSK9 promoter activity, production, and secretion, as well as the ability to affect the interaction between PCSK9 and the LDL-R in human hepatic HepG2 cells were the objects of this investigation.

## 2. Materials and Methods

### 2.1. Reagents

Eagle’s minimum essential medium (MEM) was bought from Sigma, trypsin-EDTA, penicillin, streptomycin, sodium pyruvate, non-essential amino acid solution, fetal bovine serum (FBS), plates and Petri dishes were bought from EuroClone (Milan, Italy). YVVNPDN**NEN** peptide corresponds to position 294–303 of the α subunit of β-conglycinin (UNIProtKB P0DO16), whereas YVVNPDN**DEN** peptide corresponds to positions 310–319 of the α′ subunit of β-conglycinin (UNIProtKB P11827). These peptides were certified >95% purity by HPLC (PRIMM; Milan, Italy). Simvastatin was bought from Sigma-Aldrich (Milan, Italy).

### 2.2. Cell Culture

Human hepatic cancer cell lines (HepG2) were cultured in MEM supplemented with 10% FBS, L-glutamine, sodium-pyruvate, non-essential amino acids, penicillin/streptomycin at 37 °C in a humidified atmosphere of 5% CO_2_ and 95% air.

### 2.3. qPCR

Total mRNA was extracted by using iScript reagent (Biorad, CA, USA). The cDNA was obtained by reverse-transcription (Maxima First Strand cDNA synthesis kit; Thermo Fisher, Milan, Italy). The qPCRs were performed with Thermo Maxima SYBR Green (Thermo Fisher). The sequences of the primers are reported in Table 1. The following cycling conditions were used: 95 °C, 10 min; 95 °C, 30 s and 58 °C, 1 min for 40 cycles (CFX96-Biorad). A post-PCR melt curve step was performed to confirm that a single PCR product was amplified or to detect the presence of primer-dimers or other unwanted PCR products. *ACTB* expression was determined in each sample as a normalizing gene. The mRNA levels of the genes were expressed with the relative quantity method as ^∆∆Ct^.

### 2.4. Western Blot

HepG2 cells (1.5 × 10^5^)/well were seeded in 24-well plates and treated with 350 μM of YVVNPDN**DEN** and YVVNPDN**NEN** for 24 h. The following lysis buffer was used: 40 µL of ice-cold radioimmunoprecipitation assay (RIPA) buffer + inhibitor cocktail + 1:100 PMSF (Phenylmethanesulfonyl Fluoride) + 1:100 Na-orthovanadate. Then, cells were centrifuged at 10,000 *g* for 15 min at 4 °C. The total amount of proteins was quantified by the Bicinchoninic Acid solution (BCA) assay. 30 μg of total proteins were loaded on a 10% Sodium Dodecyl Sulphate-Polyacrylamide (SDS-PAGE) gel and run at 150 V. The gels were transferred to a nitrocellulose membrane (200 V for 2 h). Target proteins, on 5% milk blocked membranes, were incubated with primary antibodies as follows: anti-PCSK9 (1:1000; Genetex, Prodotti Gianni, Milan, Italy), anti-HNF1α (1:1000; Abcam, Prodotti Gianni) and anti-β-actin (1:1000; Santacruz, DBA Italia, Milan, Italy). Secondary antibodies, conjugated with HRP, were used 1:10,000. The signal was quantified by using the Image Lab Software (Biorad). β-actin was used as the housekeeping protein.

### 2.5. Apoptosis

HepG2 cells were seeded in a 6-well plate (8 × 10^5^ cells/well) and they were treated with YVVNPDN**DEN** (350 μM), YVVNPDN**NEN** (350 μM) or simvastatin (20 μM). After 24 h, cells were washed with phosphate-buffered saline (PBS), centrifuged and stained with FITC annexin V and red-fluorescent propidium iodide (Thermo Fisher). All the samples were analysed by flow cytometry (Novocyte 3000, ACEA Bioscience, San Diego, CA, USA), measuring the fluorescence emission at 530 nm and >575 nm.

### 2.6. ELISA Assay

Conditioned media was centrifuged (13,000 rpm for 10 min.) and stored at −20 °C. The amount of PCSK9 (R&D System, Minneapolis, MN, USA) was quantified according to manufacturer’s instructions. The minimum detectable concentration was 0.219 ng/mL. Specifically, HepG2 cells were cultured in 12-well plates (6 × 10^5^ cells/well) and after 24-h treatment with YVVNPDN**DEN** peptide, YVVNPDN**NEN** peptide, and simvastatin, the medium was collected and PCSK9 dosed. Samples were not diluted, and results normalized for the total amount of proteins.

### 2.7. Transfections of Reporter Constructs

As previously described, HepG2 cells were transfected with the plasmid PCSK9 pGL3-PCSK9-D4 containing the 5′ flanking region of the PCSK9 gene from −440 to −94, relative to the ATG start codon in front of the luciferase coding sequence. The promoter constructs contain wild-type, SRE mutated (SRE-mu) and HNF1 mutated (HNF1mu) sequences [19]. To measure the human PCSK9 promoter activity, HepG2 cells were seeded in 48-well plates at a density of 4 × 10^5^ cells per well. The day after, cells were transiently transfected with pGL3-PCSK9-D4 plasmids (wild-type, SRE-mu and HNF-1-mu), with TurboFect^TM^ transfection reagent (Thermo Fisher). Twenty-four hours post transfection, cells were incubated with MEM/10%LPDS (lipoprotein deprived serum) ± YVVNPDN**DEN** (350 µM), YVVNPDN**NEN** (350 µM) and simvastatin (20 μM) for additional 24-h. Luciferase activities were measured by using Neolite reagent (Perkin Elmer, Milan, Italy) according to manufacturer’s instructions. The evaluation of β-Galactosidase activity was used to avoid treatment interference with transfection and/or protein expression. pCMV-β vector, encoding for β-galactosidase (Clontech, Mountain View, CA, USA) was transfected in HepG2 cells. Twenty-four hours after transfection cells were plated in a 48-well plate (8 × 10^4^ cells/well) and treated the next day. After 24-h treatment, cells were lysed in 130 µL of NP40 lysis buffer, diluted into reaction mix (700 µL PBS with 0.9 mM Ca and 0.5 mM Mg and 200 µL 2-Nitrophenyl β-D-galactopyranoside 2 mg/mL). The reaction was incubated at 37 °C until a faint yellow colour had developed. Reaction was stopped by adding 500 μL of 1 M Sodium Carbonate. Absorbance was read at 420 nm in Enspire multiplate reader (PerkinElmer, Waltham, MA, USA).

### 2.8. In Vitro PCSK9-LDLR Binding Assay

YVVNPDN**DEN** and YVVNPDN**NEN** peptides were tested using the in vitro PCSK9-LDLR binding assay (CycLex Co., Nagano, Japan). Plates were pre-coated with a recombinant LDLR-AB domain containing the binding site for PCSK9. Tested peptides and vehicle were added in each well of the plate and the reaction was started by adding 3 μL of His-tagged PCSK9 solution. The plate was maintained in constant shaking (300 rpm) for 2 h at room temperature. After washes, 100 μL of biotinylated anti-His-tag monoclonal antibody was added, and the plate incubated for an extra hour. HRP-conjugated streptavidin was added, and the plate incubated for 20 min. As a final step, the substrate reagent (tetra-methylbenzidine) and the stop solutions were added. The absorbance at 450 nm was measured using the Synergy H1 fluorescent plate reader (Biotek, Winooski, VT, USA).

### 2.9. Analysis of the Data

Statistical analysis was performed using the Prism statistical analysis package version 9.2 (GraphPad Software, San Diego, CA, USA). Data are given as mean ± SD of three independent experiments. Differences between treatment groups were evaluated by 1-way ANOVA or by Mann–Whitney U test. A probability value of *p* < 0.05 was considered statistically significant.

## 3. Results

### 3.1. Impact of Soy Peptides on HepG2 Cell Viability

We first evaluated whether 24-h treatment with YVVNPDN**DEN** and YVVNPDN**NEN** peptides could affect HepG2 viability. The dose of 350 µM was chosen according to previous studies on bioactivity and statin-like mechanism [11]. Staining treated cells with FITC annexin V and PI did not reveal any difference in the percentage of live cells between controls and HepG2 treated with YVVNPDN**DEN**, or YVVNPDN**NEN**. As shown in Figure 1, upon peptide treatment, green and red fluorescence did not increase, thus indicating neither a shift towards apoptosis nor necrosis. Similar conclusions were reached in the case of simvastatin, which was used as a positive control of PCSK9 activation throughout the manuscript.

### 3.2. Soy Peptides Raise PCSK9 Gene Expression and Transcriptional Activity

24-h treatment with YVVNPDN**DEN** and YVVNPDN**NEN** peptides modulated the expression of PCSK9 in HepG2 cells. The mRNA levels were raised by a non-significant +13.4% when HepG2 cells were treated with YVVNPDN**DEN** peptide and by a +90.8% (*p* < 0.05) upon YVVNPDN**NEN** peptide treatment (Figure 2A).

To evaluate whether these two peptides had a direct effect on PCSK9 transcriptional activity, HepG2 cells were transiently transfected with luciferase constructs containing the 5′-flanking region of PCSK9 gene from −440 to −94 pGL3-PCSK9-D4, relative to ATG starting codon, containing Sp1, SRE and HNF1 sites. The relative luciferase activity of the PCSK9 promoter was raised by both YVVNPDN**DEN** and YVVNPDN**NEN** peptides, respectively, by 83.7% and 86.4% (Figure 2B). This effect was maintained after the insertion of a specific mutation in SRE (+149% and +94%) and HNF1 (+93.8% and +78.9%) sites, two well-known promoter regions of PCSK9 transcription (Figure 2B). Simvastatin (20 μM) raised luciferase activity when the wildtype plasmid or the sequence containing the mutation in HNF1 was used: +142% (wildtype) and +61.7% (HNF1) (Figure 2C). As expected, simvastatin raised mRNA (Figure 2A) and in the presence of a mutation in the SRE region, simvastatin did not affect luciferase activity (Figure 2C). All data relative to luciferase activity have been normalized for β-galactosidase reporter gene, whose expression was not affected by any treatment (Appendix A).

### 3.3. Soy Peptides Decrease the Protein Levels of PCSK9 and HNF1α

Differently from mRNA levels, when the expression of PCSK9 protein was evaluated, 24-h treatment with each peptide reduced its relative expression by 21.9 ± 2% (YVVNPDN**DEN**) and 33.5 ± 12.5% (YVVNPDN**NEN**), respectively, compared with medium alone (Figure 3A). To corroborate the WB analysis, the amount of released PCSK9 was assessed. The ELISA array showed that, compared with medium alone, YVVNPDN**DEN** peptide non-significantly reduced the amount of PCSK9 released in the medium (−22.6%), whereas a statistical −42.9% fall (*p* < 0.01) was reached with YVVNPDN**NEN** peptide (Figure 3B). Simvastatin, used as a positive control, raised PCSK9 as assessed by WB and ELISA analyses (Figure 3C). To investigate further the molecular mechanisms leading to a reduced protein expression of PCSK9 upon treatment with β-conglycinin peptides, the HNF1α transcription factor was evaluated. A decrement in the protein expression of HNF1α was found only upon treatment with the YVVNPDN**NEN** peptide (Figure 3D). Simvastatin did not change the protein expression of HNF1α (Figure 3E), a finding in contrast with the results reported in hamster in which rosuvastatin increased hepatic HNF1α protein levels [20].

## 4. Discussion

Peptides from vegetable sources have been described as potentially responsible for a a cholesterol-lowering activity by statin-like mechanisms [21,22], as in the case of soy β-conglycinin peptides that lower the activity of HMGCoAR by roughly 50% [23]. Based on phage display technology, pepsin and trypsin hydrolyses, molecular docking studies and enzyme assays, the mechanism of the HMGCoAR inhibition has been postulated for some peptides, as a consequence of 3-dimensional similarity to statins [24,25].

Soy proteins, mainly derived from Glycine max, may be effectively compared, as protein sources, to animal proteins for cholesterol reduction. The intake of soy proteins in a dose-range from 25 to 100 g leads to an LDL-C reduction between 3% and 20% [26]. In particular, peptides tested in the present work are produced from the LRVPAGTTYYVVNPDNDENLRMIT (UNIProtKB: P11827) fragment of soybean βCG. YVVNPDN**DEN** and YVVNPDN**NEN** inhibit the HMGCoAR activity and increase the protein expression of both SREBP2 and LDLR receptor [11]. Differently from other soybean peptides, e.g., IAVPTGVA, IAVPGEVA, and LPYP (from glycinin hydrolysis with pepsin) which are less effective inhibitors of HMGCoAR activity (IC_50_ values equal to 247, 222, and 300 μM [10]), YVVNPDN**DEN** and YVVNPDN**NEN** peptides (from β-conglycinin) reduce the enzyme activity with IC_50_ values equal, respectively, to 150 and 200 μM [11]. However, the impact of bioavailable YVVNPDN**DEN** and YVVNPDN**NEN** peptides on PCSK9 has not been investigated. In the present study, we have shown that part of the effect of these two peptides on the expression of LDL-R could be mediated by the decrement in the protein expression of PCSK9, one of the main regulators of LDL-R recycling [27].

YVVNPDN**NEN** was the peptide with the strongest impact on the expression of PCSK9 as assessed by quantity of protein secreted in the medium of HepG2 cells (−41%) if compared with YVVNPDN**DEN** (−22%). Similar conclusions were reached when PCSK9 intracellular levels were evaluated. These findings are discordant with PCSK9 gene expression that was raised, mostly by YVVNPDN**NEN**. It is tempting to speculate that YVVNPDN**DEN** and YVVNPDN**NEN** may inhibit the auto-processing of PCSK9 by interacting with its catalytic domain, thus reducing both the intracellular and extracellular levels of PCSK9 [28,29]; in response to these events, the activation of SREBP-mediated transcriptional activities of PCSK9 can be expected.

The impact of soy peptides on PCSK9 has also been found for lunasin, which downregulates the expression of PCSK9, thus leading to a raised expression of LDLR at transcriptional and translational levels, at least in HepG2 cells [30]. In order to understand the mechanism by which YVVNPDN**DEN** and YVVNPDN**NEN** could modulate the expression of PCSK9, we evaluated the transcriptional activity upon treatment with YVVNPDN**DEN** and YVVNPDN**NEN** peptides. Considering that PCSK9 transcription is controlled via cis regulatory elements embedded in the proximal promoter region of *PCSK9* gene where the Sp1, SRE and HNF1 sites are located, a PCSK9 promoter luciferase reporter plasmid, containing mutations for both SRE and HNF responsive elements, has been used [31]. Exposure of HepG2 hepatocytes to YVVNPDN**DEN** and YVVNPDN**NEN** increases the transcriptional activity of the PCSK9 promoter. Neither construct variant containing mutations in the SRE motif or in the HNF1 motif changed the transcriptional activity mediated by YVVNPDN**DEN** and YVVNPDN**NEN** peptides. These latter findings would imply the involvement of additional transcriptional factors, e.g., signal transducer and activator of transcription-3 (STAT3) [19] and forkhead box class O 3 (FOXO3) [32] or post-transcriptional mechanisms involved in the protein expression of PCSK9. Considering that *PCSK9* is the first identified gene which is involved in cellular cholesterol metabolism that utilizes an HNF1 binding site to cooperate with SRE, we also evaluated the impact of YVVNPDN**DEN** and YVVNPDN**NEN** peptides on the protein expression of HNF1α. Under our experimental conditions, we found a downregulation of HNF1α, thus excluding its involvement in the activation of PCSK9 promoter activity. Although the liaison between HNF1α and PCSK9 has been previously described for lupin peptides GQEQSHQDEGVIVR and LILPKHSDAD [33,34], these studies did not explore the impact on PCSK9 transcriptional activity.

Finally, we did not find any effect of YVVNPDN**DEN** and YVVNPDN**NEN** peptides in the PCSK9-LDLR binding (Appendix A), a mechanism found in the case of the lupin peptides GQEQSHQDEGVIVR and LILPKHSDAD [35].

## 5. Conclusions

Although sharing a common effect on LDL-R levels through the inhibition of HMGCoAR activity [11], only the YVVNPDN**NEN** has an additional mechanism via the downregulation of PCSK9 intracellular levels and its secreted amount. The present inhibitory lipid-lowering pathway of soy which is related to the LDLR up-regulation could be also mediated by a PCSK9 antagonism. This evidence could be of major interest leading to new approaches for cardiovascular prevention.

## Figures and Tables

**Figure 1 nutrients-14-00193-f001:**
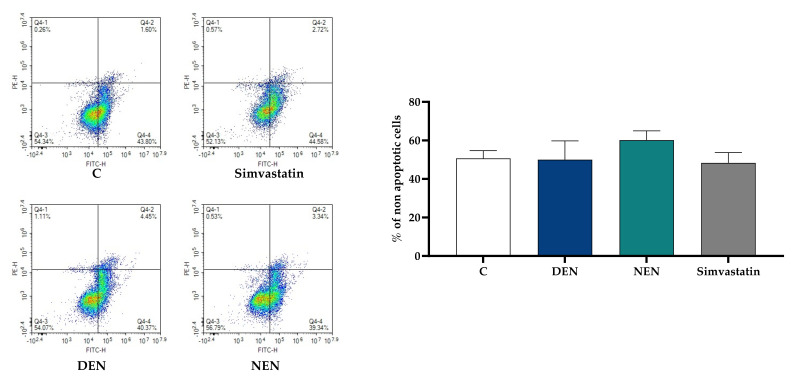
Impact of soy peptides on HepG2 cell viability. HepG2 cells were incubated with YVVNPDN**DEN** (350 μM) and YVVNPDN**NEN** (350 μM) in MEM supplemented with 10% FBS for 24 h. Subsequently, they were stained with FITC annexin V and red-fluorescent propidium iodide. Simvastatin (20 μM) was tested since it was used as a positive control of PCSK9 activation throughout the manuscript. At least three independent experiments were conducted. PCSK9, proprotein convertase subtilisin/kexin type 9; C, control (medium alone). DEN stands for YVVNPDN**DEN** and NEN stands for YVVNPDN**NEN**.

**Figure 2 nutrients-14-00193-f002:**
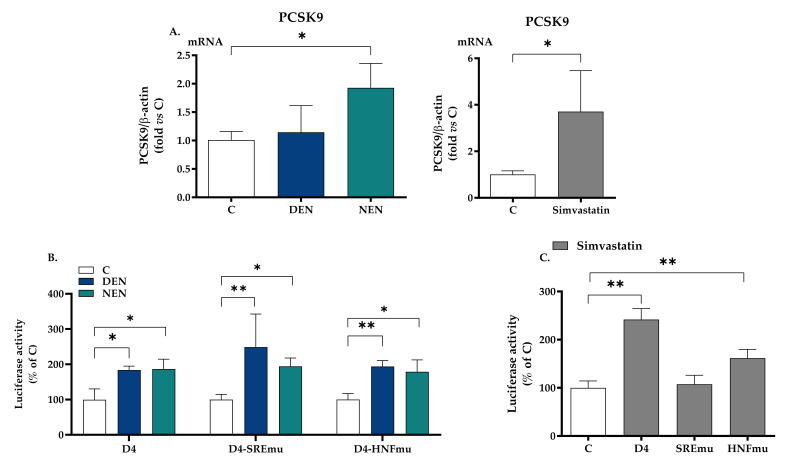
Impact of soy peptides on PCSK9 gene expression and transcriptional activity. Panel (**A**) shows the gene expression of PCSK9 in HepG2 cells incubated with YVVNPDN**DEN** (350 μM) and YVVNPDN**NEN** (350 μM) in MEM supplemented with 10% FBS for 24 h. Simvastatin (20 μM) was used as a positive control. At the end of the incubation, total RNA was extracted. Panels (**B**,**C**) show HepG2 cells transfected with pGL3-PCSK9-D4, pGL3-PCSK9-SREmut, and pGL3-PCSK9-HNF1mut. The day after the transfection, the medium was replaced by MEM containing 10% LPDS supplemented with YVVNPDN**DEN** (350 μM) and YVVNPDN**NEN** (350 μM) and the specific luciferase activity was determined. Panel (**C**) shows luciferase activity upon treatment with simvastatin. At least three independent experiments were conducted. * *p* < 0.05 versus control (ANOVA) and ** *p* < 0.01 versus control (ANOVA). C, control (medium alone). LPDS, lipoprotein deprived serum; HNF1, hepatocyte nuclear factor-1; PCSK9, proprotein convertase subtilisin/kexin type 9; SRE, sterol regulatory element. Simvastatin has been used as a positive control (**C**). DEN stands for YVVNPDN**DEN** and NEN stands for YVVNPDN**NEN**.

**Figure 3 nutrients-14-00193-f003:**
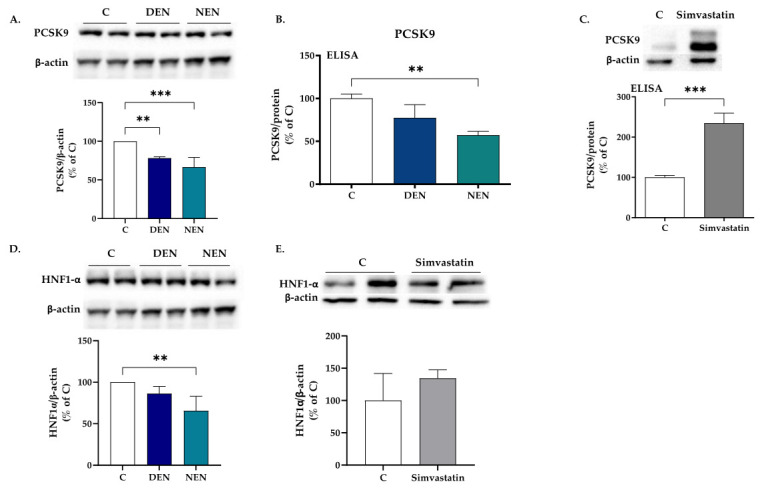
Impact of soy peptides on PCSK9 and HNF1α protein expression. HepG2 cells were seeded in MEM 10% LPDS. After 48 h, the medium was replaced with MEM with 10% LPDS supplemented with YVVNPDN**DEN** (350 μM) and YVVNPDN**NEN** (350 μM). After 24 h, expression of PCSK9 (**A**, **B**) and HNF1α was evaluated (**D**). HNF1α protein expression upon 24-h treatment with 20 μM simvastatin (**E**). β-actin was used as housekeeping protein. PCSK9 levels were assessed by ELISA (data were normalized for the total amount of proteins; panel (**B**). At least three independent experiments were conducted. ** *p* < 0.01 versus control (ANOVA); *** *p* < 0.001 versus control (ANOVA) (**A**); *** *p* < 0.001 versus control (Mann-Whitney) (**C**). LPDS, lipoprotein deprived serum. C, control (medium alone); DEN stands for YVVNPDN**DEN**; NEN stands for YVVNPDN**NEN**.

**Table 1 nutrients-14-00193-t001:** Primer sequences and efficiency.

Gene	Forward	Reverse	Efficiency
*PCSK9*	5′-CCTGCGCGTGCTCAACT-3′	5′-GCTGGCTTTTCCGAAACTC-3′	105%
*ACTB*	5′-TTCTACAATGAGCTGCGTGTG-3′	5′-GGGGTGTTGAAGGTCTCAAA-3′	95%

A, adenine; C, cytosine; T, thymine; G, guanine; ACTB, β-actin; PCSK9, Proprotein convertase subtilisin/kexin 9.

## Data Availability

Data will be available at request.

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
