# Peer review of "Impact of Soy β-Conglycinin Peptides on PCSK9 Protein Expression in HepG2 Cells"

_nutrients, 2021, doi:10.3390/nu14010193_

Round 1
Reviewer 1 Report
The authors characterized the effect of two soy peptides, β-conglycinin-derived YVVNPDNDEN and YVVNPDNNEN on the expression of proprotein convertase subtilisin/kexin type 9 (PCSK9). The authors have taken a good effort to present their results clearly, but Some issues should be addressed by authors.
Abstract:
- YVVNPDNNEN raised PCSK9 transcriptional activity, not correct. It has the lowest PCSK9 protein expression.
- Neither peptide inhibited the PCSK9:LDLR interaction. No data support it, please add the experimental data.
- Conclusions: Although sharing a common effect on LDL-R levels through the inhibition of 3-hydroxy-3-methylglutaryl CoA reductase activity, …. This part is out of this research experiment.
- Please revise the abstract.
- 281-282, These findings are discordant with PCSK9 gene expression that was raised, mostly by YVVNPDNNEN. Please discuss it in detail.
L 283, as recently demonstrated, there is a hepatic sensing loop that regulates PCSK9 secretion; PCSK9 re-entry to the liver via LDLR triggers a sensing loop regulating PCSK9 secretion [27]. This sentence is puzzling. Please give the expression clearly.
Author Response
Reviewer#1
The authors characterized the effect of two soy peptides, β-conglycinin-derived YVVNPDNDEN and YVVNPDNNEN on the expression of proprotein convertase subtilisin/kexin type 9 (PCSK9). The authors have taken a good effort to present their results clearly, but Some issues should be addressed by authors.
Abstract:
- YVVNPDNNEN raised PCSK9 transcriptional activity, not correct. It has the lowest PCSK9 protein expression.
We thank the reviewer for pointing this out. The sentence has been re-wrote for clarity and now reads as follows “Treatment with YVVNPDNNEN resulted in a rise in PCSK9 gene expression, PCSK9 transcriptional activity and a decrement in intracellular and secreted protein levels. YVVNPDNNEN reduced the protein expression of HNF1a.”
- Neither peptide inhibited the PCSK9:LDLR interaction. No data support it, please add the experimental data.
Data have been added as a supplementary Figure 2
- Conclusions: Although sharing a common effect on LDL-R levels through the inhibition of 3-hydroxy-3-methylglutaryl CoA reductase activity, …. This part is out of this research experiment.
Data on 3-hydroxy-3-methylglutaryl CoA reductase activity have been previously demonstrated by Lammi et al., and our sentence meant to highlight a further lipid-lowering effect driven by these peptides besides the reduction of 3-hydroxy-3-methylglutaryl CoA reductase.
- Please revise the abstract.
The abstract was revised
- 281-282, These findings are discordant with PCSK9 gene expression that was raised, mostly by YVVNPDNNEN. Please discuss it in detail.
We thank the reviewer for point out this issue. The discussion was re-written as follows:
“These findings are discordant with PCSK9 gene expression that was raised, mostly by YVVNPDNNEN. It is tempting to speculate that YVVNPDNDEN and YVVNPDNNEN may inhibit the auto-processing of PCSK9 by interacting with its catalytic domain, thus reducing both the intracellular and extracellular levels of PCSK9 [28,29]; in response to these events, the activation of SREBP-mediated transcriptional activities of PCSK9 can be expected.”
“These latter findings would imply the involvement of additional transcriptional factors, e.g., signal transducer and activator of transcription-3 (STAT3) [19] and forkhead box class O 3 (FOXO3) [32] or post-transcriptional mechanisms involved in the protein expression of PCSK9.”
“Under our experimental conditions, we found a downregulation of HNF-1α transcription factor, thus excluding its involvement in the activation of PCSK9 promoter activity. Although the liaison between HNF-1α and PCSK9 has been previously described for lupin peptides GQEQSHQDEGVIVR and LILPKHSDAD [33,34], these studies did not explore the impact on PCSK9 transcriptional activity.”
L 283, as recently demonstrated, there is a hepatic sensing loop that regulates PCSK9 secretion; PCSK9 re-entry to the liver via LDLR triggers a sensing loop regulating PCSK9 secretion [27]. This sentence is puzzling. Please give the expression clearly.
The sentence has been erased. This was referred to the recent manuscript by Fazio showing that PCSK9 re-entry to the liver via LDLR triggers a sensing loop regulating PCSK9 secretion. PCSK9i therapy enhances the secretion of PCSK9, an effect that contributes to the increased plasma PCSK9 levels in treated subjects (J Am Coll Cardiol. 2021 Oct 5;78(14):1437-1449).
Reviewer 2 Report
Well-structured article with a pertinent approach to the level of LDL-C reduction with a protective effect on cardiovascular diseases.
I suggest completing information in line 99 with addition of time regarding the length of the PCR.
The study at the molecular level made it possible to advance in the mechanism of regulation of LDL-R. It was noticed that only the YVVNPDNNEN peptide interferes with the expression of the pro-protein PCSK9, which regulates LDL-R.
Well-structured article with adequate methodology, good level of English.
Recommendations: line 99 needs to complete information about the PCR, namely the cycle extension time.
Author Response
Well-structured article with a pertinent approach to the level of LDL-C reduction with a protective effect on cardiovascular diseases.
I suggest completing information in line 99 with addition of time regarding the length of the PCR.
We thank the reviewer for her/his thoughtful comment. The info requested have been added.
The study at the molecular level made it possible to advance in the mechanism of regulation of LDL-R. It was noticed that only the YVVNPDNNEN peptide interferes with the expression of the pro-protein PCSK9, which regulates LDL-R.
Well-structured article with adequate methodology, good level of English.
Recommendations: line 99 needs to complete information about the PCR, namely the cycle extension time.
The following sentence has been added “The analysis was performed with the CFX96 (Biorad) with the following cycling conditions: 95°C, 10min; 95°C, 30 sec and 58°C, 1 min for 40 cycles. A post-PCR melt curve step was performed to confirm that a single PCR product was amplified or to detect the presence of primer-dimers or other unwanted PCR products.”